# Advancing tools for simulation-based inference

Henning Bahl[1], Victor Bresó-Pla[1], Giovanni De Crescenzo[1] and Tilman Plehn[1,2]

**1** Institut für Theoretische Physik, Universität Heidelberg, Germany
**2** Interdisciplinary Center for Scientific Computing (IWR),
Universität Heidelberg, Germany

## Abstract

We study the benefit of modern simulation-based inference to constrain particle interactions at the LHC. We explore ways to incorporate known physics structures into likelihood estimation, specifically morphing-aware estimation and derivative learning. Technically, we introduce a new and more efficient smearing algorithm, illustrate how uncertainties can be approximated through repulsive ensembles, and show how equivariant networks can improve likelihood estimation. After illustrating these aspects for a toy model, we target di-boson production at the LHC and find that our improvements significantly increase numerical control and stability.



## 1  Introduction

Since the Higgs discovery, the experimental and theoretical communities have turned the LHC into the first precision hadron collider in history. Measurements and simulations of LHC event kinematic patterns are high-dimensional. This is an advantage because a lot of information can be used to test the underlying theory. However, it challenges traditional methods, which rely on at most few-dimensional histograms as summary statistics. Modern machine learning techniques can be used to overcome this bottleneck [1, 2], specifically unbinned likelihood (ratio) estimation [3].

To interpret LHC measurements systematically, we need to (i) optimally extract information from observed and simulated events, and (ii) publish this information in a useful manner. In both cases, total or fiducial rate measurements alone are not enough. Instead, we need to analyze and publish full likelihoods or likelihood ratios, including an accurate modeling of systematic uncertainties and their correlations. One approach to optimal inference is the so-called matrix element method, enabled by modern machine learning [4, 5]. Alternatively, high-dimensional and unbinned likelihood ratio estimation can be achieved using simulation-based inference (SBI), which is also based on modern machine learning [6–9]. SBI was first implemented for LHC purposes in the public MADMINER tool [10]. The promise of this analysis approach has been documented by several phenomenological studies, for instance targeting $VH$ [11], $ZH$ [12], $t\bar{t}$ [12], and Higgs pair production [13] using the effective theory extension of the Standard Model (SMEFT) [14], or CP-phases in $t\bar{t}H$ [15,16] and $WH$ production [17].

One outcome of these phenomenological analyses is a set of open questions concerning the numerical stability of likelihood ratio estimation, especially its scaling towards the fully exclusive phase space and high-dimensional theory space. Here, we can use our physics understanding of LHC phase space and of the perturbative structure of the underlying quantum field theory.

This is for instance evident in SMEFT, a theoretical description which allows us to combine rate and kinematic information from different processes [18–20]. In SMEFT, the scattering matrix element is expressed as a truncated Taylor series in the Wilson coefficients. This means that we can extract first and second derivatives rather than the full likelihood ratio [12,21,22], including systematic uncertainties [23]. Alternatively, this structure can also be exploited by learning the likelihood ratio at a set of basis points [6,24].

In this paper, we target some of the known numerical shortcomings of the standard MAD-MINER tool box. In Sec. 2 we discuss different ways of using our physics knowledge about LHC scattering amplitudes to improve likelihood learning. The advantages of morphing-aware sampling and derivative learning are best illustrated for a toy model in Sec. 3.1. The fractional smearing technique introduced in Sec. 3.2 leads to a critical numerical improvement for our toy model and also for the $WZ$ application discussed in detail in Sec. 4. The benefits from the covariant L-GATr architecture become relevant once we probe higher-dimensional phase space, for instance for $WZ$ production at the reconstruction level.

## 2 Learning the likelihood

The extraction of likelihood ratios at the LHC involves two phase spaces and their underlying distributions, $z_p$ for the hard scattering and $x$ for instance for the (smeared) reconstruction level. Our goal is to extract the likelihood ratio $r(x|\theta, \theta_0)$, defined as the ratio between the likelihood with free $\theta$ against a fixed reference hypothesis $\theta_0$,

$$r(x|\theta, \theta_0) \equiv \frac{p(x|\theta)}{p(x|\theta_0)}. \tag{1}$$

There are two main strategies which can be used to achieve this. First of all, we review the structure of the problem.

**LHC likelihood factorization**

Following Refs. [6–8], we can assume that our parameters of interest only affect the hard scattering observables $z_p$, while the parton shower (transferring the $z_p$ to the shower variables $z_s$), the detector simulation (transferring the $z_s$ to the detector variables $z_d$), and the forming of the reco-level observables $x$ out of the $z_d$ are independent of $\theta$. In this case, we assume that the likelihood factorizes as

$$p(x|\theta) = \int dz_d dz_s dz_p p(x|z_d) p(z_d|z_s) p(z_s|z_p) p(z_p|\theta) = \int dz_p p(x|z_p) p(z_p|\theta), \tag{2}$$

where in the last step $p(z_p|\theta)$ can be evaluated using parton-level event generators:

$$p(z_p|\theta) = \frac{1}{\sigma(\theta)} \frac{d\sigma(z_p|\theta)}{dz_p}. \tag{3}$$

The differential cross-section reads

$$d\sigma(z_p|\theta) = (2\pi)^4 \int dx_1 dx_2 d\Phi \frac{f_1(x_1, Q^2) f_2(x_2, Q^2)}{2x_1 x_2 s} |\mathcal{M}(z_p|\theta)|^2, \tag{4}$$

where $f_i(x, Q^2)$ are the parton densities depending on the momentum transfer $Q$ and partonic momentum fractions $x_i$, $\mathcal{M}(z_p|\theta)$ is the matrix element of the hard scattering and $d\Phi$ is the phase-space element. This factorization is important because it allows us to compute numerically the *joint likelihood ratio*, which is simply the ratio of the joint probabilities. The full likelihood ratio is clearly intractable, but the joint likelihood ratio simply reduces to the parton level likelihood ratio:

$$r(x, z_d, z_s, z_p|\theta, \theta_0) := \frac{p(x|z_d) p(z_d|z_s) p(z_s|z_p) p(z_p|\theta)}{p(x|z_d) p(z_d|z_s) p(z_s|z_p) p(z_p|\theta_0)} = \frac{p(z_p|\theta)}{p(z_p|\theta_0)} =: r(z_p|\theta, \theta_0). \tag{5}$$

This quantity is available from simulation and it can be exploited to make training easier, given we are feeding the NN with more information. We can now present the two main approaches to approximate the quantity of interest $r(x|\theta, \theta_0)$.

**Likelihood learning via classifier**

By training a discriminator $D(x)$ to separate the effects of the parameter choices $\theta$ and the reference hypothesis $\theta_0$, we can extract the likelihood ratio at the reconstruction level. If the classifier is optimally trained, it has the form

$$D_{\text{opt}}(x|\theta, \theta_0) = \frac{p(x|\theta_0)}{p(x|\theta_0) + p(x|\theta)}. \tag{6}$$

Then, the likelihood ratio is given by

$$r(x|\theta, \theta_0) = \frac{1 - D_{\text{opt}}(x|\theta, \theta_0)}{D_{\text{opt}}(x|\theta, \theta_0)} \,. \tag{7}$$

This is the CARL method implemented in MADMINER [7,10]. To learn the classifier we define a learnable $f_\varphi(x|\theta)$ and study the functional

$$F[f_\varphi] = \int d\theta q(\theta) \left[ \int dx\, p(x|\theta_0) \log(f_\varphi(x|\theta)) + \int dx\, p(x|\theta) \log(1 - f_\varphi(x|\theta)) \right], \tag{8}$$

where we ignore the second argument $\theta_0$. Minimizing $F$ with respect to $f_\varphi(x|\theta)$ leads to the condition

$$0 = \frac{\delta F}{\delta f_\varphi} \propto \frac{p(x|\theta_0)}{f_\varphi(x|\theta)} - \frac{p(x|\theta)}{1 - f_\varphi(x|\theta)} \quad \Leftrightarrow \quad f_\varphi(x|\theta) = \frac{p(x|\theta)}{p(x|\theta_0) + p(x|\theta)} \,. \tag{9}$$

This method does not make use of parton-level information. To include it, the functional form can be adapted leading to the same minimum for $f_\varphi(x|\theta)$

$$F[f_\varphi] = -\int d\theta q(\theta) \left[ \int dx \int dz_p\, p(x|z_p) p(z_p|\theta_0) \frac{1}{1 + r(z_p|\theta, \theta_0)} \log f_\varphi(x|\theta) \right. \tag{10}$$
$$\left. + \int dx \int dz_p p(x|z_p) p(z_p|\theta) \left(1 - \frac{1}{1 + r(z_p|\theta, \theta_0)}\right) \log(1 - f_\varphi(x|\theta)) \right].$$

This is the ALICE method implemented in MADMINER [25]. In principle, it should lead to faster training given the additional parton-level information the network is fed. The ALICE method is implemented through the loss

$$\mathcal{L} = -\left\langle \left[ \frac{1}{1 + r(z_p|\theta, \theta_0)} \log r_\varphi(x|\theta, \theta_0) \right.\right.$$
$$\left.\left. + \left(1 - \frac{1}{1 + r(z_p|\theta, \theta_0)}\right) \log(1 - r_\varphi(x|\theta, \theta_0)) \right] \right\rangle_{x,z_p \sim \frac{p(x|z_p)p(z_p|\theta) + p(x|z_p)p(z_p|\theta_0)}{2}; \theta \sim q(\theta)} \,. \tag{11}$$

**Likelihood regression**

As an alternative, we can also regress to the reco-level likelihood using parton-level or hard-scattering information [6]. While, in general, we cannot evaluate the integral in Eq.(2), we can learn it by using $p(z_p|\theta)$. For our learnable $f_\varphi(x|\theta)$, we investigate the functional

$$F[f_\varphi] = \int dx \int d\theta \int dz_p q(\theta) p(x|z_p) p(z_p|\theta) \left[ f(z_p|\theta) - f_\varphi(x|\theta) \right]^2, \tag{12}$$

where $f(z_p)$ is some function of the parton-level variables. Minimizing $F$ with respect to $f_\varphi(x|\theta)$ leads to the condition,

$$0 = \frac{\delta F}{\delta f_\varphi(x|\theta)} \propto \int dz_p p(x|z_p) p(z_p|\theta) \left[ f(z_p|\theta) - f_\varphi(x|\theta) \right]. \tag{13}$$

Here, the functional derivative with respect to $f_\varphi(x|\theta)$ effectively removes the integrals over $\theta$ and $x$. Solving for $f_\varphi(x|\theta)$, we obtain

$$f_\varphi(x|\theta) = \frac{\int dz_p p(x|z_p) p(z_p|\theta) f(z_p|\theta)}{\int dz_p p(x|z_p) p(z_p|\theta)} = \frac{\int dz_p p(x|z_p) p(z_p|\theta) f(z_p|\theta)}{p(x|\theta)} \,. \tag{14}$$

We can use this method for the joint likelihood ratio, such that

$$f(z_p|\theta) \equiv r(z_p|\theta_0, \theta) = \frac{1}{r(z_p|\theta, \theta_0)} = \frac{p(z_p|\theta_0)}{p(z_p|\theta)}, \tag{15}$$

and

$$f_\varphi(x|\theta) \equiv \frac{1}{r_\varphi(x|\theta, \theta_0)} \approx \frac{1}{r(x|\theta, \theta_0)}. \tag{16}$$

This way we learn the likelihood ratio at the reco-level. This minimization can be implemented through the loss

$$\mathcal{L} = \left\langle \left[ r(z_p|\theta, \theta_0) - r_\varphi(x|\theta, \theta_0) \right]^2 \right\rangle_{x, z_p \sim p(x|z_p)p(z_p|\theta); \theta \sim q(\theta)}. \tag{17}$$

Each training event includes the parton-level momenta $z_{p,i}$, the reco-level observables $x_i$, and the theory parameters $\theta_i$. In addition, we attach the parton-level likelihood ratio as a label. As we will see in Sec. 2.2, one can regress on a range of possible targets using this MSE, all relying on the knowledge of the parton-level likelihood ratio.

The exact form of the loss is important. For example, if taking a different exponent or computing the mean-squared-error w.r.t. $\log r(z_p|\theta, \theta_0)$, $r_\varphi(x)$ will in general not converge to $r(x|\theta, \theta_0)$. Linear operations, like scaling the targets to have mean of zero and a standard deviation of one, will not affect the convergence, since multiplication and addition commute with the integral. This can be seen by focusing on Eq. (14): we can see that $f(z_p|\theta)$ needs to appear linearly in the integral in order for the cancellation between numerator and denominator $p(z|\theta)$ to happen (see Eq. (15)).

The most straightforward way to learn the $\theta$-dependence of $r$ is to sample the training data from various $\theta$-hypotheses and to condition the neural network on $\theta$. Since directly sampling from a large set of theory hypotheses is not feasible in practice, the samples must be generated using morphing techniques, as discussed below. Alternatively, the theory parameter dependence can be directly imprinted by splitting the likelihood ratio into several functions approximated by separate neural networks.

## 2.1 Morphing

All modern Monte Carlo generators combine phase-space sampling with additional dimensions, for instance helicities or color. Naturally, we can do the same with model parameters to produce an efficient training dataset to learn likelihood ratios [6]. For instance, for BSM physics described by an effective Lagrangian up to dimension six,

$$\mathcal{L}_{\text{SMEFT}} = \mathcal{L}_{\text{SM}} + \sum_i \frac{c_i}{\Lambda^2} O_i \equiv \mathcal{L}_{\text{SM}} + \sum_i \theta_i O_i, \tag{18}$$

the differential cross-section for the hard scattering has the form

$$|\mathcal{M}(z_p|\theta)|^2 = |\mathcal{M}_{\text{SM}}(z_p)|^2 + \theta_i |\mathcal{M}_i(z_p)|^2 + \theta_i \theta_j |\mathcal{M}_{ij}(z_p)|^2. \tag{19}$$

Even though we use SMEFT as our example, the same form applies to any polynomial dependence on couplings of interest. This form also holds beyond leading order.

Using Eq.(19), we can factorize the differential and total partonic cross-sections as

$$\frac{d\sigma(z_p|\theta)}{dz_p} \equiv w_a(\theta) f_a(z_p), \qquad \text{and} \qquad \sigma(\theta) = w_a(\theta) \int dz_p f_a(z_p). \tag{20}$$

This separates the $\theta$-dependence in $w_a(\theta)$ and the $z_p$-dependence in $f_a(z_p)$. For a single $\theta$, the sum over $a$ includes three terms, the constant SM term, the linear, and the quadratic BSM contributions. The normalized probability distribution is then

$$p(z_p|\theta) = \frac{1}{\sigma(\theta)} \frac{d\sigma(z_p|\theta)}{dz_p} = \frac{w_a(\theta)}{\sigma(\theta)} f_a(z_p). \tag{21}$$

For a suitable set of points $\theta_i$ and a set of $a$-terms, this matrix relation can be inverted

$$p(z_p|\theta_i) = \frac{w_a(\theta_i)}{\sigma(\theta_i)} f_a(z_p) \qquad \Leftrightarrow \qquad f_a(z_p) = \left[\frac{w_a(\theta_i)}{\sigma(\theta_i)}\right]^{-1} p(z_p|\theta_i). \tag{22}$$

This way, we can use basis densities $p(z_p|\theta_i)$ to reconstruct the full dependence on $\theta$:

$$p(z_p|\theta) = \frac{w_a(\theta)}{\sigma(\theta)} \left[\frac{w_a(\theta_i)}{\sigma(\theta_i)}\right]^{-1} p(z_p|\theta_i). \tag{23}$$

This equation can be understood as a vector-matrix-vector multiplication

$$p(z_p|\theta) = \vec{v}(\theta) M^{-1}(\theta) \vec{p}(z_p), \qquad \text{with} \qquad M_{ai} = \frac{w_a(\theta_i)}{\sigma(\theta_i)}. \tag{24}$$

This morphing procedure does not only work for the parton-level densities but also for the reco-level densities, following from Eq.(2). It can be exploited in different ways: first, we can use it to generate training samples throughout the $\theta$-space to train a neural network conditioned on $\theta$; alternatively, we can perform morphing-aware likelihood estimation.

**Morphing-aware likelihood estimation**

The morphing structure of Eq.(24) can be transferred to the reco-level likelihood ratio

$$r(x|\theta, \theta_0) = \vec{v}(\theta) M^{-1}(\theta) \vec{r}(x), \qquad \text{with} \qquad r_i(x) = r(x|\theta_i, \theta_0). \tag{25}$$

Using the BCE loss in Eq.(11) and sampling only from $\theta_i$ and $\theta_0$, we first train networks to approximate $\vec{r}$. These are then combined using Eq.(25) to obtain the complete likelihood ratio. This way, the $\theta$-dependence is not learned but directly imposed.

The morphing-aware likelihood-ratio estimation in MADMINER works slightly differently. Instead of learning the $r_i$ with individual losses, the learned $\vec{r}_\varphi$ are combined into $r_\varphi$ for which the MSE loss is calculated. Consequently, the neural networks have to be trained simultaneously using data covering $x$-space and $\theta$-space. The sampling of $\theta$-space induces instabilities [6] and the morphing of the trained networks for the likelihood ratio is an additional source of uncertainty. Moreover, there is a high degeneracy in the scalar product of $\vec{v}M^{-1}$ with $\vec{r}$ making it hard to regress on $\vec{r}$. As we will demonstrate in Sec. 3, learning the likelihood ratio at every benchmark point leads to more stable results, at least for low-dimensional problems. For higher-dimensional problems, a suitable choice of the morphing basis points becomes very important to ensure numerical stability.

While background contributions can in principle be handled as an additional contribution to $|\mathcal{M}_{\text{SM}}(z_p)|^2$, it is advantageous for the network training to handle these separately via an additional classifier network. This is discussed in App. D.

## 2.2 Derivative learning

While morphing-aware likelihood learning relies on a set of benchmark points, we can also use the structure of Eq.(19). Following Refs. [21,22], we expand the unnormalized reco-level likelihood ratio

$$R(x|\theta,\theta_0) \equiv \frac{d\sigma(x|\theta)/dx}{d\sigma(x|\theta_0)/dx} = \frac{\sigma(\theta)p(x|\theta)}{\sigma(\theta_0)p(x|\theta_0)}, \tag{26}$$

around $\theta_0$ as

$$R(x|\theta,\theta_0) = 1 + (\theta-\theta_0)_i R_i(x) + (\theta-\theta_0)_i (\theta-\theta_0)_j R_{ij}(x). \tag{27}$$

The first and second derivatives

$$R_i(x) \equiv \frac{\partial}{\partial \theta_i} R(x|\theta,\theta_0)\bigg|_{\theta=\theta_0}, \qquad \text{and} \qquad R_{ij}(x) \equiv \frac{\partial^2}{\partial \theta_i \partial \theta_j} R(x|\theta,\theta_0)\bigg|_{\theta=\theta_0}, \tag{28}$$

do not depend on $\theta$. Eq. (27) is exact if $R(x|\theta,\theta_0)$ is quadratic in $\theta$, as it is the case for dimension-six SMEFT analyses.

As derived in Eq.(14), we can learn the derivatives of $R(x|\theta,\theta_0)$ using a MSE loss. Using the parton-level joint derivatives

$$f(z_p|\theta) \equiv f(z_p) = R_i(z_p) \equiv \frac{\partial}{\partial \theta_i} \frac{d\sigma(z_p|\theta)/dz_p}{d\sigma(z_p|\theta_0)/dz_p}\bigg|_{\theta=\theta_0} = \frac{\partial_{\theta_i}|\mathcal{M}(z_p|\theta)|^2}{|\mathcal{M}(z_p|\theta_0)|^2}\bigg|_{\theta_0},$$

$$f(z_p|\theta) \equiv f(z_p) = R_{ij}(z_p) \equiv \frac{\partial^2}{\partial \theta_i \partial \theta_j} \frac{d\sigma(z_p|\theta)/dz_p}{d\sigma(z_p|\theta_0)/dz_p}\bigg|_{\theta=\theta_0} = \frac{\partial_{\theta_i}\partial_{\theta_j}|\mathcal{M}(z_p|\theta)|^2}{|\mathcal{M}(z_p|\theta)|^2}\bigg|_{\theta_0}, \tag{29}$$

and sampling only from $\theta_0$, the learned functions will converge to

$$f_\varphi(x|\theta) \equiv R_{\varphi,i}(x) \approx R_i(x), \qquad \text{and} \qquad f_\varphi(x|\theta) \equiv R_{\varphi,ij}(x) \approx R_{ij}(x). \tag{30}$$

The deduction proceeds in the same way as with the likelihood ratio $r$, since the derivatives with respect to the theory parameters can be pulled out of the integrals.

With this construction, we only need to simulate events at one point in $\theta$-space, preferentially the SM point now assumed to be $\theta_0 = 0$. Exploiting Eq.(19), we then calculate $R_i(z_p)$ and $R_{ij}(z_p)$, which can be obtained from the simulator for the SM sample. After learning the reco-level $R_{i,ij}(x)$, we obtain the likelihood ratio via

$$r(x|\theta,\theta_0) = \frac{\sigma(\theta_0)}{\sigma(\theta)} \frac{d\sigma(x|\theta)/dx}{d\sigma(x|\theta_0)/dx} = \frac{\sigma(\theta_0)}{\sigma(\theta)} R(x|\theta,\theta_0). \tag{31}$$

The ratio of total cross-sections can be treated analogously to Eq.(27),

$$\sigma(\theta) = \sigma(\theta_0) + (\theta-\theta_0)_i \sigma_i(\theta_0) + (\theta-\theta_0)_i (\theta-\theta_0)_j \sigma_{ij}(\theta_0), \tag{32}$$

with

$$\sigma_i(\theta_0) \equiv \frac{\partial}{\partial \theta_i} \sigma(\theta)\bigg|_{\theta=\theta_0}, \qquad \text{and} \qquad \sigma_{ij}(\theta_0) \equiv \frac{\partial^2}{\partial \theta_i \partial \theta_j} \sigma(\theta)\bigg|_{\theta=\theta_0}. \tag{33}$$

These derivatives can be obtained by summing the parton-level $R_{i,ij}(z_p)$ or the estimated $R_{\varphi,i}(x)$ and $R_{\varphi,ij}(x)$ over the event sample. While the first option is more precise, the second results in a more stable behaviour for small $\theta$. This can be understood by expanding the summed log-likelihood

$$\sum_i^N \log r(x_i|\theta,\theta_0) \simeq \left[\sum_i^N R_{\varphi,j}(x_i) - N \frac{\sigma_j}{\sigma(\theta_0)}\right] (\theta_j - \theta_{0,j}) + \mathcal{O}\big((\theta_j - \theta_{0,j})^2\big)$$

$$= \mathcal{O}\big((\theta_j - \theta_{0,j})^2\big). \tag{34}$$

If the $\sigma_j$ are calculated by summing $R_j(z_p)$ and $R_{\varphi,j}(x)$ are not learned perfectly, the first-order term will not completely vanish for $\theta$ close to $\theta_0$. Calculating the derivatives by summing $R_{\varphi,i}(x)$ and $R_{\varphi,ij}(x)$ enforces a vanishing first-order term by construction.

It is important to note that this derivative-learning method fails if the weights change significantly within the considered $\theta$-range in some $x$-region. In that case a region with very large weights will be undersampled, leading to numerical instabilities in the most sensitive phase space regions.

## 2.3 Fractional smearing

When simulating reco-level events $x$ from the parton level $z_p$, all 4-momenta are smeared out by parton shower and detector effects. A unit-weight event at the parton level turns into a unit-weight event at the reco-level. To estimate the changing phase-space densities in sparsely populated regions more efficiently, we can describe the smearing using a set of weighted events, given that the network training can trivially be extended to this case [26].

To this end, we pass some events through the simulation chain multiple times, for instance events with large parton-level likelihood ratio or with large parton-level derivatives. These events contribute to the loss function with a reduced weight, to ensure that the event sample still follows the distribution $p(x|z_p)p(z_p|\theta)$.

Let us assume that we want to learn the target function $r(z_p|\theta, \theta_0)$, omitting the index in $\theta_i$. We follow a series of steps manipulating the parton-level events:

1. calculate the mean $\mu$ and standard deviation $\sigma$ of the target $r(z_p|\theta, \theta_0)$;

2. assign a fractional-smearing weight $w$ to every event, initially set to one;

3. smear an event by copying it $n$ times and assigning $w = 1/n$ to each copy;

4. define a threshold, for instance smear all events with $|w \times r(z_p|\theta, \theta_0) - \mu| > t\sigma$ for a given $t$;

5. smear until the threshold value is reached.

This sample of smeared and weighted parton level events is passed through the simulation. The fractional-smearing weights are incorporated into the MSE loss, for example for the likelihood ratio regression for a fixed $\theta$,

$$\mathcal{L} = \left\langle \left[ r(z_p|\theta, \theta_0) - r_{\varphi}(x|\theta) \right]^2 \right\rangle_{x, z_p \sim p(x|z_p)p(z_p|\theta)} = \sum_i^N w_i \left[ r(z_{p,i}|\theta, \theta_0) - r_{\varphi}(x_i|\theta) \right]^2, \quad (35)$$

illustrating the role of the weights $w$. The sum runs over the event sample. If we apply cuts to the reco-level events, only some of the fractionally-smeared events belonging to the same parton-level event might survive. This is not a problem, since the cut event sample still corresponds to a weighted sample of the distribution $p(x|z_p)p(z_p|\theta)$.

As an alternative to using fractional smearing for preparing the neural-network training dataset, boosted-decision trees which learn to "bin" the phase space can be used. As has been demonstrated in Refs. [21, 22] for the derivative learning approach, the division of the phase space allows for accurate estimates even in sparsely populated phase-space regions.

## 2.4 L-GATr

To improve the likelihood learning we can employ equivariant neural networks, which have been shown to enhance the performance for different problems in particle physics [27–30].

These models encode knowledge about the spacetime symmetry directly into their structure. This way, the network does not need to spend training resources in learning the symmetry properties of the data, and its operations get restricted to those allowed by the symmetry. This makes equivariant networks fast to train, resistant to overfitting, and sample-efficient.

We use the Lorentz-Equivariant Geometric Algebra Transformer (L-GATr) [30–32] for likelihood learning. L-GATr is a transformer-based model that processes data in the spacetime geometric algebra representation. To construct it, a geometric product is introduced to the vector space to create higher-order objects from the original Lorentz-vectors. This operation modifies the properties of the basis of the vector space starting from the relation

$$\{\gamma^\mu, \gamma^\nu\} = 2g^{\mu\nu}. \tag{36}$$

This anti-commutation relation defines gamma matrices and establishes a close connection between the real spacetime algebra and the complex Dirac algebra. Inspired by this connection, we build multivectors forming the spacetime algebra:

$$x = x^S \, 1 + x^V_\mu \, \gamma^\mu + x^B_{\mu\nu} \, \sigma^{\mu\nu} + x^A_\mu \, \gamma^\mu\gamma^5 + x^P \, \gamma^5, \qquad \text{with} \qquad \begin{pmatrix} x^S \\ x^V_\mu \\ x^B_{\mu\nu} \\ x^A_\mu \\ x^P \end{pmatrix} \in \mathbb{R}^{16}. \tag{37}$$

A multivector consists of 16 components organized in grades according to the length of $\gamma^\mu$-matrix products needed to express them. Specifically, $x^S 1$ represents a scalar, $x^V_\mu \gamma^\mu$ a vector, $x^B_{\mu\nu}\sigma^{\mu\nu}$ a geometric bilinear, $x^A_\mu\gamma^\mu\gamma^5$ an axial vector, and $x^P\gamma^5$ a pseudoscalar.

The spacetime algebra not only expresses a wide range of objects in Minkowski space, it also offers a way to define learnable equivariant transformations on particle physics data. A transformation $f$ is equivariant with respect to Lorentz transformations $\Lambda$ if

$$f\big(\Lambda(x)\big) = \Lambda\big(f(x)\big). \tag{38}$$

We encode this condition in L-GATr by considering the fact that every grade transforms under a given representation of the Lorentz group. This structure of the geometric algebra allows us to easily develop equivariant versions of any network architecture. L-GATr performs this adaptation using transformer structures, with equivariant versions of linear, attention, layer-normalization, and activation operations,

$$\bar{x} = \text{LayerNorm}(x),$$
$$\text{AttentionBlock}(x) = \text{Linear} \circ \text{Attention}(\text{Linear}(\bar{x}), \text{Linear}(\bar{x}), \text{Linear}(\bar{x})) + x,$$
$$\text{MLPBlock}(x) = \text{Linear} \circ \text{Activation} \circ \text{Linear} \circ \text{GP}(\text{Linear}(\bar{x}), \text{Linear}(\bar{x})) + x, \tag{39}$$
$$\text{Block}(x) = \text{MLPBlock} \circ \text{AttentionBlock}(x),$$
$$\text{L-GATr}(x) = \text{Linear} \circ \text{Block} \circ \text{Block} \circ \cdots \circ \text{Block} \circ \text{Linear}(x).$$

We also include the MLPBlock containing a geometric product operation to maximize network expressivity. Further details are provided in Refs. [30–32].

For likelihood learning our data consists of particle properties organized as tokens. For the L-GATr input we embed the 4-momenta $p^\mu$ as the vector grade in the algebra,

$$x^V_\mu = p_\mu, \qquad \text{and} \qquad x^S = x^T_{\mu\nu} = x^A_\mu = x^P = 0. \tag{40}$$

In addition to this geometric vector, our data also contains particle identification and partial kinematic input, like missing transverse momentum. They are given to the network as independent scalar inputs and are evaluated by the transformer in parallel to the multivectors.

Both tracks are mixed in the equivariant linear layers. As L-GATr output we select the scalar component of the corresponding multivectors. In our case, we perform this operation on a global token, an extra particle object that is appended to the rest of the particle tokens in each sample. This extra token is empty at the input level and gains meaning at the output level through the network training [32].

# 3 Toy model

To illustrate the ideas behind our SBI tools we use a toy model with two Gaussians at parton level,

$$p_\alpha(z_p|\theta) = \frac{\mathcal{N}_{0,1}(z_p) + \theta^2 \mathcal{N}_{\alpha,0.1}(z_p)}{1 + \theta^2}, \tag{41}$$

where $\theta$ stands for the theory parameter we want to constrain. The parameter $\alpha$ controls the distance between the two Gaussians. We then add a simple Gaussian smearing,

$$p(x|z_p) = \mathcal{N}_{0,0.7}(z_p), \tag{42}$$

emulating parton shower and detector. $x$ represents a reco-level observable. For this toy model, the reco-level likelihood becomes

$$p_\alpha(x|\theta) = \frac{\mathcal{N}_{0,1.22}(x) + \theta^2 \mathcal{N}_{\alpha,0.71}(x)}{1 + \theta^2}. \tag{43}$$

Then, the likelihood ratio is given by

$$r_\alpha(x|\theta, \theta_0) = \frac{1 + \theta_0^2}{1 + \theta^2} \frac{\mathcal{N}_{0,1.22}(x) + \theta^2 \mathcal{N}_{\alpha,0.71}(x)}{\mathcal{N}_{0,1.22}(x) + \theta_0^2 \mathcal{N}_{\alpha,0.71}(x)}. \tag{44}$$

In addition to the reduced dimensionality, the main simplifications of this toy model is that we know the normalization exactly and that we do not consider any continuum backgrounds.

## 3.1 Morphing-aware vs. derivative learning

First, the Gaussian toy model allows us to compare morphing-aware and derivative likelihood estimation. The locality in phase space is determined by $\alpha$. For $\alpha \sim 1$ — i.e., if it is similar

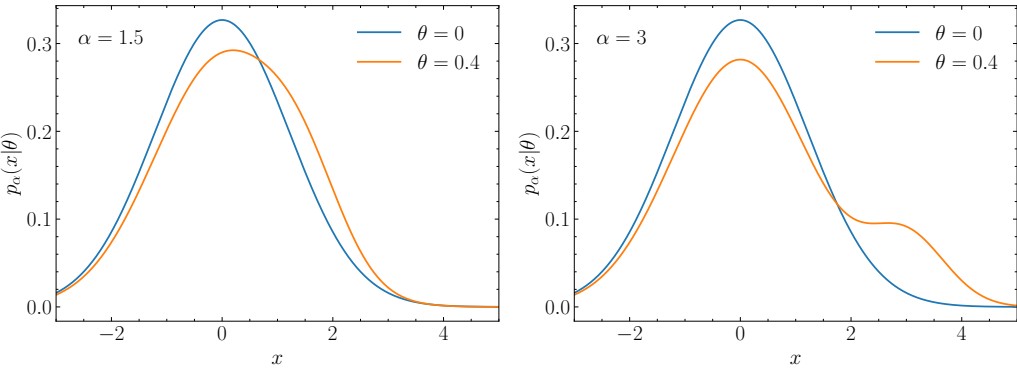

Figure 1: Likelihoods for local and non-local cases. Both show the full $p(x|\theta)$ for $\theta = 0$ and $\theta = 0.4$.

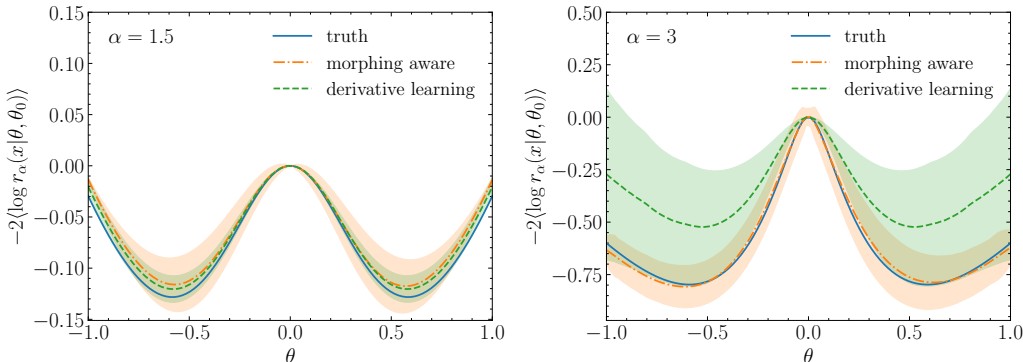

Figure 2: Comparison between the likelihood ratios estimated using the morphing-aware and derivative learning approaches for $\alpha = 1.5$ (left) and $\alpha = 3$ (right). The truth value is $\theta = 0.6$.

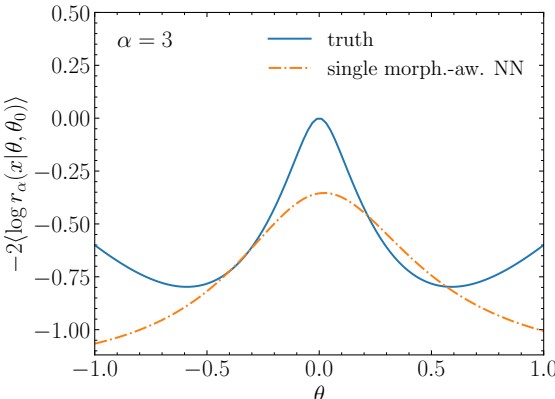

Figure 3: Likelihood ratio estimated using the single neural network morphing-aware implementation of Ref. [6], compared to the truth.

to the widths of the Gaussians —, the likelihood will hardly vary when we change $\theta$ away from $\theta = 0$. On the other hand, for $\alpha \gg 1$ even small changes in $\theta$ will shift the bulk of the likelihood, as illustrated for $\alpha = 1.5$ and $\alpha = 3$ in Fig. 1. This means that for $\alpha \sim 1$ we can sample from the $\theta = 0$-hypothesis, while for $\alpha \gg 1$ we need to sample from different basis points using a morphing-aware approach.

To test derivative learning, we generate $3 \times 10^5$ events at the reference point $\theta_0 = 0$. For morphing-aware sampling, we generate $10^5$ events at $\theta = -1, 0, 1$. For both, we use the same number of networks with the same amount of parameters and training epochs. Each network is implemented as a repulsive ensemble [2, 33, 34], to estimate the uncertainty due to limited training data. For testing the likelihood ratio estimates, we use data generated with $\theta = 0.6$.

The results of the toy model reflect two aspects of the likelihood ratio learning problem. In the left panel of Fig. 2 we see that for $\alpha = 1.5$ the two methods have comparable performance and can be trusted. For $\alpha = 3$, the derivative learning method gives worse results. The training data at the $\theta = 0$ hypothesis does not cover the relevant phase-space regions, as we can already see in Fig. 1. Finally, the repulsive ensembles give us an indication on how reliable each method is, for instance comparing the uncertainty bands for $\alpha = 3$.

For our toy model, morphing-aware sampling [6] provides an accurate likelihood ratio estimation. As explained in Sec. 2.1, the novelty in our implementation is that we train the likelihood ratios for each basis point independently. We can compare our implementation with the version from Ref. [6], again with the same size of the training dataset and the same number of network parameters and training epochs. For the $\theta$-prior in Eq.(17), we use a standard Gaussian. We only check the non-local case, because this is where morphing-aware sampling has advantages. The results are shown in Fig. 3, indicating that for the toy model our implementation is more stable.

The stability of our morphing-aware likelihood estimation can, however, be compromised for more than one theory parameter. In this case, the choice of basis points becomes very important, and a non-optimal choice will lead to large coefficients in Eq.(25) from the matrix inversion. This, in turn, results in enhanced uncertainties due to imperfect network training.

### 3.2 Fractional smearing

To illustrate our novel fractional smearing, we use the same toy model as before in an even less local setting with $\alpha = 4$. This models a SMEFT situation, where kinematic tails can be particularly sensitive to the theory parameters. We use the more challenging derivative learning and generate $10^5$ training events for $\theta = 0$. This way, the large-$x$ region, which has the largest sensitivity to $\theta$ and a large derivative $R_{\theta^2}$, is only sparsely populated.

Figure 4 shows the distribution of the reco-level second derivative with respect to $\theta$, for the original and fractionally-smeared datasets. The original dataset only features a few isolated events with large $R_{\theta^2}$. The underlying distribution is hard to learn. The fractionally-smeared dataset has a much more balanced distribution, which is far easier to learn. Additional visualizations of the datasets are provided in App. B.

Next, we show the learned second derivatives as a function of $x$ in the left panel of Fig. 5. The single network and the repulsive ensemble trained on the fractional-smearing dataset capture the large-$x$ behaviour of the truth much better than the neural network trained on the original dataset. Finally, we show the expected log-likelihood ratios in the right panel of Fig. 5. Again, the networks trained on the fractionally-smeared dataset are much closer to the true likelihood ratio, illustrating the significantly improved training.

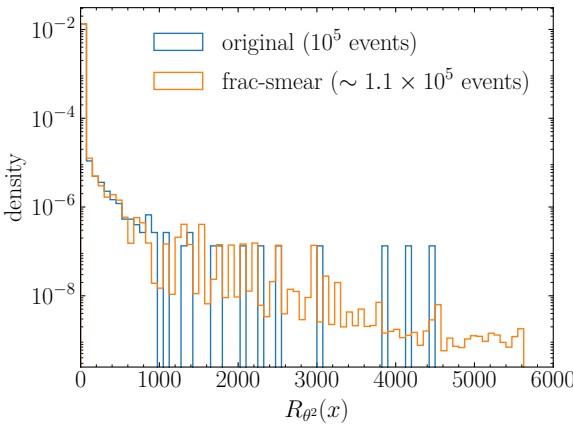

Figure 4: Histogram of the reco-level second derivative $R_{\theta^2}$, for the original dataset (blue) and the fractionally-smeared dataset (orange).

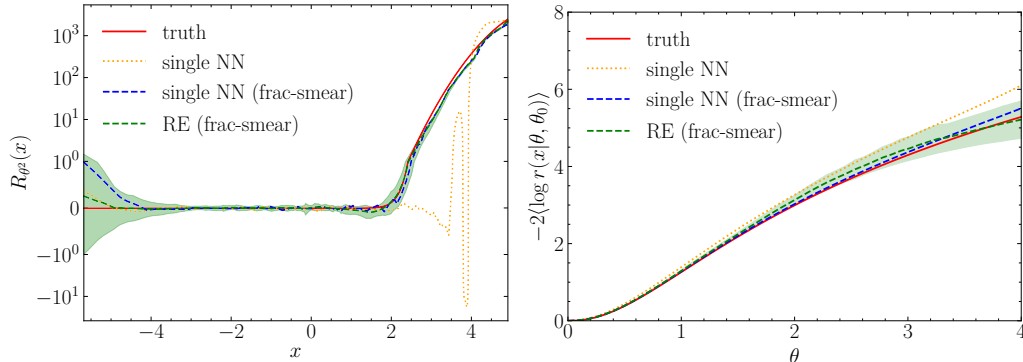

Figure 5: Left: learned second derivative as a function $x$ for various setups compared to the truth. Right: expectation value of the learned likelihood ratio as a function of $\theta$ compared to the true likelihood ratio.

# 4 LHC application: $pp \to W^{\pm}Z$

As an LHC physics example, we use $W^{\pm}Z$ production at Run 3, with $\sqrt{s} = 13.6\,\text{TeV}$ and $\mathcal{L} = 300\,\text{fb}^{-1}$. At tree level, this process has two contributions, distinguished by the $Z$-coupling either to quarks or to the $W$-boson. We modify the SM-interactions through the dimension-6 operators

$$
\begin{aligned}
\mathcal{O}_{\Phi WB} &= \Phi^{\dagger} \tau^a \Phi W^a_{\mu\nu} B^{\mu\nu}\,, \\
\mathcal{O}_{WWW} &= \epsilon^{abc} W^{a\nu}_{\mu} W^{b\rho}_{\nu} W^{c\mu}_{\rho}\,, \\
\mathcal{O}^{(3)}_{\Phi q} &= (\Phi^{\dagger} i \overleftrightarrow{D^a}_{\mu} \Phi)(\bar{Q}_L \tau^a \gamma^{\mu} Q_L)\,.
\end{aligned}
\tag{45}
$$

Here, $\Phi$ is the Higgs doublet, $W^{\mu\nu}$ and $B^{\mu\nu}$ are the $SU(2)_L$ and $U(1)_Y$ field strengths, $D_{\mu}$ is the covariant derivative, and $Q_L$ the left-handed quark doublets. The UV cutoff $\Lambda$ is set to 1 TeV for our numerical analysis. In the $WZ$ process, the first operator modifies the $Z$-couplings to quarks and the $WWZ$-coupling. The second modifies only the $WWZ$-coupling; the third

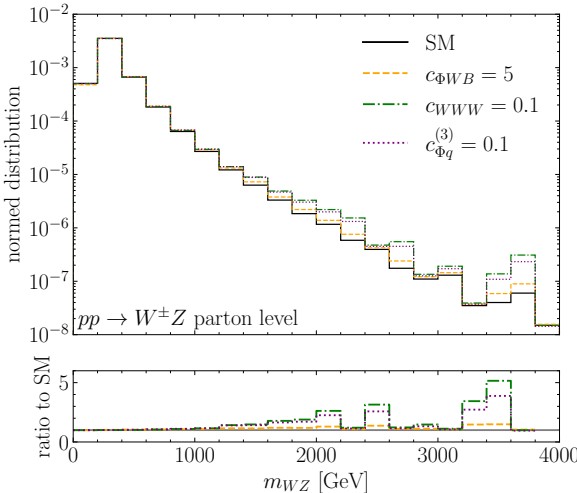

Figure 6: Normalized distribution of the $WZ$ invariant mass for the SM and three Wilson coefficients, one at a time.

one, only the $Z$ coupling to quarks. The corresponding Wilson coefficients are our theory parameters,

$$\theta = \left(c_{\Phi WB}, c_{WWW}, c_{\Phi q}^{(3)}\right), \qquad \text{with} \qquad \theta_0 = (0, 0, 0). \tag{46}$$

We use MADGRAPH5_AMC@NLO 3.5.0 [35] for event generation at the leading order, employing the SMEFTATNLO [36] UFO model. For the reco-level analysis, we decay the vector bosons leptonically using MADSPIN [37]. The parton shower is PYTHIA8 8.306 [38], the detector simulation DELPHES 3.5.0 [39], and the jet algorithm FASTJET 3.3.4 [40].

The NLO QCD corrections have a sizeable dependence on the phase space region and the entering Wilson coefficients [41]. This dependence is, however, reduced by applying a jet veto — as done below for our reco-level analysis. Therefore, we approximate the NLO QCD corrections by multiplying the total rate with an NLO $K$-factor [42].

We illustrate the effect of each Wilson coefficient on the distribution of the $WZ$-invariant mass in Fig. 6. The effects of $c_{WWW}$ and $c_{\Phi q}^{(3)}$ on the high-energy tail are specially visible.

## 4.1 Parton level

At parton level, we apply derivative learning at the SM point to learn the likelihood ratio employing repulsive ensembles. We tested that at parton level L-GATr yields only marginal improvements. This is due to the simplicity of the task, there is not much to be gained from increasing the sophistication of the network. Moreover, we also show results using the morphing-aware approach.

We compute the parton-level derivatives and likelihood ratios using the built-in reweighting functionality of MADGRAPH5_AMC@NLO. It is important to reweight to the full amplitude and not only the amplitude corresponding to the helicity of each event, as it is done by default in MADGRAPH5_AMC@NLO — see Refs. [43, 44] for more details.

As input features for the parton-level training, we use the Mandelstam variables $s$ and $t$, together with the charge of the final-state $W$. We evaluate the likelihood ratio using data samples generated at the SM point and at the exemplary BSM point $c_{\Phi WB} = 1, c_{WWW} = c_{\Phi q}^{(3)} = 0.1$. We train the derivative networks at the SM point $\theta_0$ using a training dataset containing 300k events and a validation dataset using 100k events. For morphing-aware learning, each training dataset is composed of 250k SM points and 250k BSM points. The choice of morphing basis points is detailed in App. C. The SM and BSM test datasets contain 100k and 300k events,

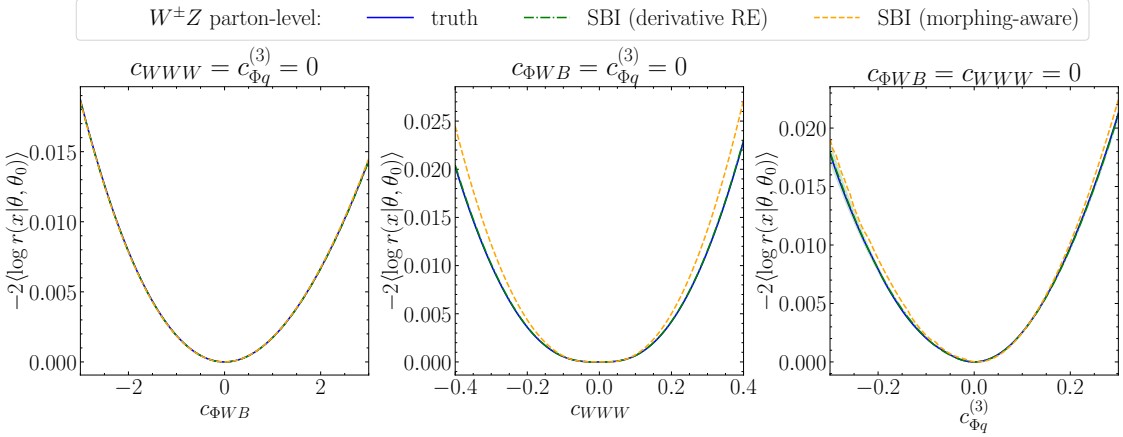

Figure 7: Log-likelihood ratio averaged over the event sample, showing the truth, the morphing-aware SBI result, and the derivative learning SBI result using repulsive ensembles as a function of $c_{\Phi WB}$ (left), of $c_{WWW}$ (center), and of $c_{\Phi q}^{(3)}$ (right).

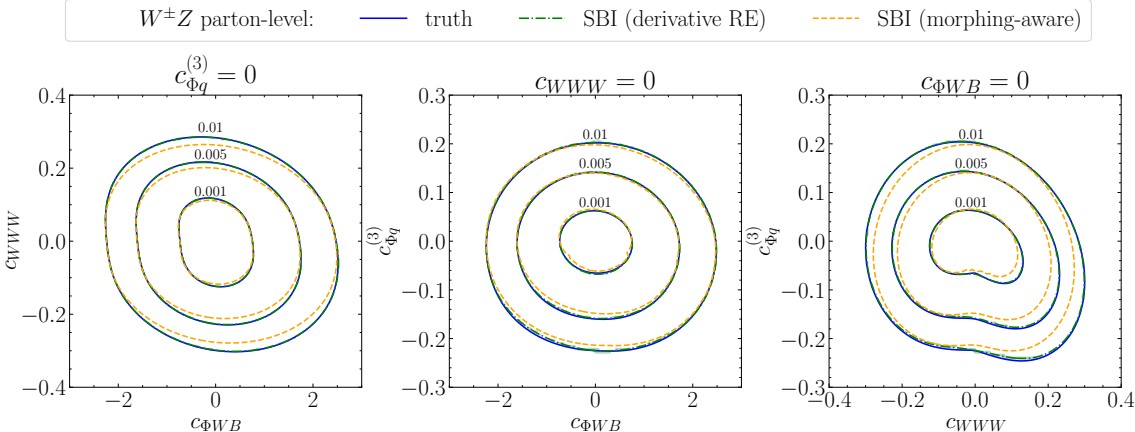

Figure 8: Contours of the log-likelihood ratio averaged over the event sample showing the truth, the morphing-aware SBI result, and the derivative learning SBI result using repulsive ensembles as a function of $c_{\Phi WB}$ and $c_{WWW}$ (left), of $c_{\Phi WB}$ and $c_{\Phi q}^{(3)}$ (center), and of $c_{WWW}$ and $c_{\Phi q}^{(3)}$ (right).

respectively. For derivative learning, we scale the input as well as output features to mean zero and standard deviation one.

First, we look at the average log-likelihood ratio using the SM sample. We show the results of the derivative learning and the morphing-aware approach for a single non-zero Wilson coefficient in Fig. 7. For the derivative learning approach, the differences to the true log-likelihood ratio are orders of magnitude smaller than the overall log-likelihood, and the uncertainty estimate using repulsive ensembles is correspondingly small. This reflects that the training dataset and the evaluation dataset are generated at the same $\theta$-point. The morhping-aware approach performs worse but still provides a good estimation of the true likelihood ratio. The largest deviations from the true log-likelihood ratio appear across the $c_{WWW}$ dimension. A possible explanation — as can be inferred by the flat shape close to zero of the true curve — could be that the quadratic SMEFT operator insertion dominates over the linear one, complicating the choice of suitable morphing basis points.

Similarly, the 2-dimensional correlations from derivative learning, displayed in Fig. 8, are in excellent agreement with the true log-likelihood ratio. The derivative learning uncertainty estimates from the repulsive ensembles are again very small. The morhping-aware approach again performs worse.

Next, we evaluate the average log-likelihood ratio for the BSM event sample in Fig. 9. The agreement with the true likelihood ratio for the derivative learning is very good. Even though the parameter point is well outside the expected $3\sigma$ exclusion region of LHC Run-3 (see below) and features a sizeable enhanced tail in the $m_{WZ}^T$ distribution, the derivative learning yields stable results. The morphing-aware approach provides results close to the true likelihood ratio but sizeable deviations are visible.

We conclude that the considered scenario is local enough in the observable space for derivative learning to perform very well (see Sec. 3.1). The better phase-space coverage of the morphing-aware approach does not provide any benefit for the considered scenario.

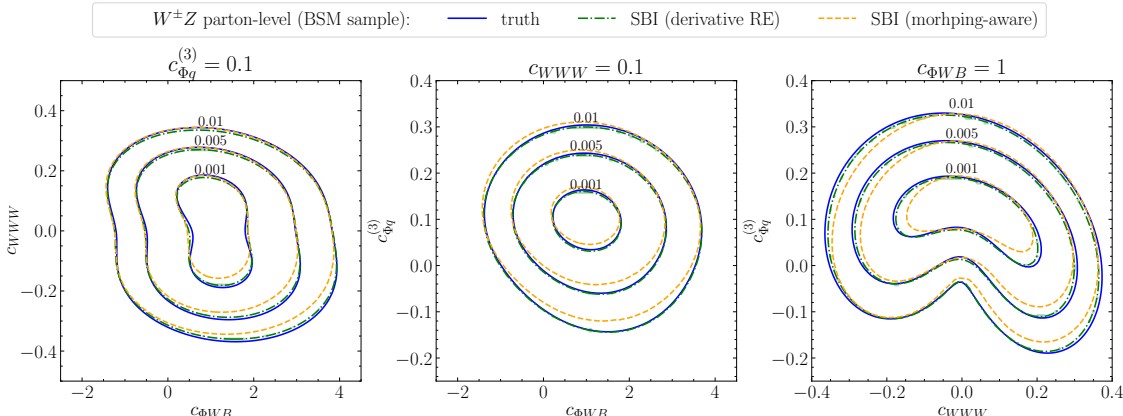

Figure 9: Contours of the log-likelihood ratio averaged over the BSM event sample showing the truth, the morphing-aware SBI result, and the derivative learning SBI result using repulsive ensembles as a function of $c_{\Phi WB}$ and $c_{WWW}$ (left), of $c_{\Phi WB}$ and $c_{\Phi q}^{(3)}$ (center), and of $c_{WWW}$ and $c_{\Phi q}^{(3)}$ (right).

## 4.2 Reconstruction level

For the reco-level analysis, we consider the leptonic $W$ and $Z$-decays and require exactly three leptons and no jet with

$$ p_{T,\ell} > 15 \text{ GeV}, \qquad |\eta_\ell| < 2.5, \qquad p_{T,j} > 20 \text{ GeV}, \qquad |\eta_j| < 2.5. \qquad (47) $$

Before the likelihood inference, we impose the pre-selection cuts

$$ m_Z^{\ell\ell} = 81.2 \ldots 101.2 \text{ GeV}, \qquad m_T^W > 30 \text{ GeV}, \qquad \not{E}_T > 45 \text{ GeV}, \qquad p_T^{\ell W} > 20 \text{ GeV}. \qquad (48) $$

They ensure that the backgrounds are negligible for the likelihood inference. As shown in App. D, incorporating backgrounds into the likelihood inference is straightforward.

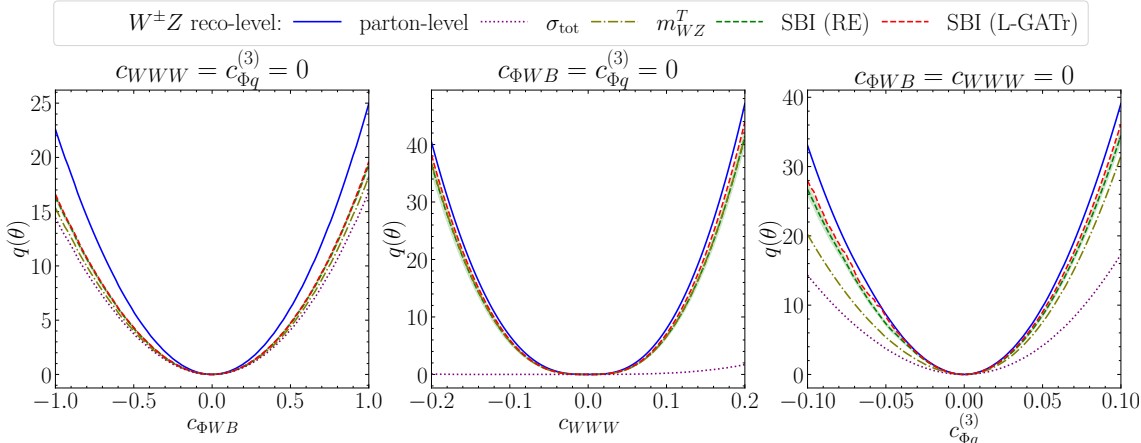

Figure 10: Test-statistics $q(\theta)$ for $\mathcal{L} = 300\text{fb}^{-1}$ showing the parton-level truth, the reco-level SBI result using repulsive ensembles, and L-GATr. For comparison, we also show results based only on the cross-section and $m_{WZ}^T$. All methods are evaluated as functions of $c_{\Phi WB}$ (left), of $c_{WWW}$ (center), and of $c_{\Phi q}^{(3)}$ (right).

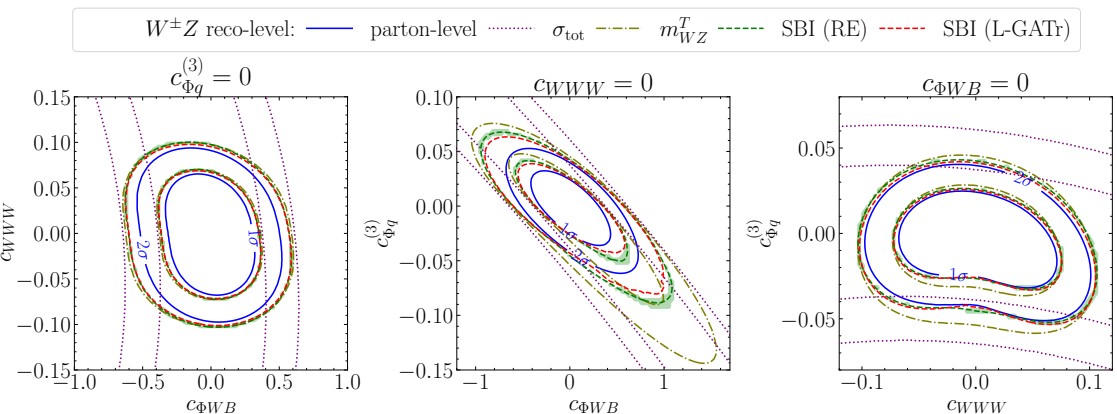

Figure 11: Negative log-likelihood for $\mathcal{L} = 300\text{fb}^{-1}$ showing the parton-level truth, the reco-level SBI result using repulsive ensembles, and L-GATr. For comparison, we also show results based only on the cross-section and $m_{WZ}^T$. All methods are evaluated as a functions of $c_{\Phi WB}$ and $c_{WWW}$ (left), of $c_{\Phi WB}$ and $c_{\Phi q}^{(3)}$ (center), and of $c_{WWW}$ and $c_{\Phi q}^{(3)}$ (right).

For the reconstruction level we focus on derivative learning. As seen at parton level, it yields reliable results for our $WZ$ analysis. This method is well motivated for our current analysis, but we want to point out that it could be insufficient for a full NLO analysis. The reason is that the phase space and theory parameter dependence can cause the appearance of large weights in underpopulated regions. In such a scenario, morphing aware likelihood learning could be a more stable method. Additionally, fractional smearing would be a useful tool to further tackle this issue.

For our $WZ$ reco-level analysis, we employ L-GATr in addition to repulsive-ensemble MLP results. The main input features are the three lepton 4-momenta and $\not{E}_T$. In addition, we provide the sum of the lepton charges, the number of jets, and several reconstructed high-level observables: $m_Z^{\ell\ell}$, the reconstructed $Z$ boson transverse momentum $p_T^Z$, $p_T^{W\ell}$, $m_T^W$, and the transverse mass of the di-boson system $m_T^{WZ}$.

To generate the training and validation datasets, we employ fractional smearing with the threshold $t = 0.5$ for each of the targets $R_{i,ij}(z_p)$. The training dataset contains around 650k events, the validation dataset 220k events, and the test dataset (generated without fractional smearing), 200k events.

Fig. 10 shows the results at reco-level for each of the three Wilson coefficients. In it, we replace the average likelihood ratio with the test statistics $q(\theta)$ derived on the basis of the full likelihood, see App. A. For reference, we also show the limits from the cross-section only and from the 1-dimensional $m_{WZ}^T$ histogram with seven bins and the boundaries 200 GeV, 400 GeV, 600 GeV, 800 GeV, 1 TeV, 1.5 TeV, and 2.5 TeV. As expected, the parton-level truth is the most constraining input. The parton shower, the detector resolution, and the presence of missing energy result in a loss information. At reco-level, limits from the total rate are poor, in particular for $c_{WWW}$, which mainly affects the tail of the $m_{WZ}^T$ distribution. Including kinematic information via the $m_{WZ}^T$-histogram tightens the expected limits significantly. Using SBI, the limits are further improved, most notably for $c_{\Phi q}^{(3)}$. We again find a very small uncertainty estimate from the repulsive ensemble, indicating the robustness of our results. L-GATr leads to a further improvement for $c_{WWW}$ and $c_{\Phi q}^{(3)}$.

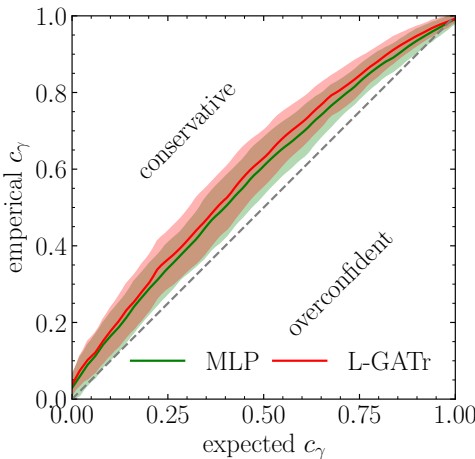

Figure 12: Expected coverage versus empirical coverage obtained scanning the likelihood in the full 3-dimensional $\theta$-space. The colored bands indicate the $1\sigma$ uncertainty in the calculation of the empirical coverage.

The 2-dimensional $1\sigma$ and $2\sigma$ confidence regions are shown in Fig. 11. The number of events is given by the expected SM events at Run-3, with $\mathcal{L} = 300\text{fb}^{-1}$. The cross-section-only limits have flat directions which are lifted by including kinematic information. As expected, the histogram information is the least constraining, followed by the repulsive ensemble SBI results. The most constraining power at reco-level comes from L-GATr, with a substantial improvement over the MLP along the diagonal in the $(c_{\Phi WB}, c_{\Phi q}^{(3)})$ direction.

Finally, we show in Fig. 12 the empirical coverage versus the expected coverage, described in App. A. To evaluate the empirical coverage, we generate random samples from the SM test dataset with the same event count as the SM prediction. For each of these samples, we evaluate the likelihood over the 3-dimensional $\theta$-space. We repeat this procedure 500 times to determine the empirical coverage and its statistical uncertainty. We see that our learned likelihoods are slightly conservative, meaning that the true SM point is within the confidence region more often than expected. This shows that our trained likelihood is not overestimating the sensitivity on the theory parameters. Although it is not necessary in the present case, the training loss of the neural networks can be adapted to ensure that the likelihood estimation is not overconfident [45].

# 5 Outlook

Simulation-based inference is the lead theme of modern LHC analyses, especially in view of the tenfold increase of data at the HL-LHC. It allows us to extract optimal information by comparing observed and simulated data without any low-dimensional or binned information bottleneck. To apply it systematically to LHC data, we need to employ modern machine learning, not only to improve the forward simulations, but also to get access to the unbinned likelihood (ratio).

In this paper we have targeted the likelihood ratio extraction and improved some of the standard methods and techniques [10]. First, we have compared different ways to encode the perturbative physics structure into the likelihood learning and shown how uncertainties due to limited training data can be estimated using repulsive ensembles. A critical numerical improvement came from fractional smearing and weighted network training. To further improve

the training towards complex scattering processes, we have employed a Lorentz-equivariant geometric algebra transformer, L-GATr. We have illustrated and studied all these technical improvements for a simple toy model, including a detailed comparison of the advantages and disadvantages of morphing-aware and derivative learning.

For a physics application, we have looked at determining three Wilson coefficients simultaneously from $W^{\pm}Z$ production kinematics. At parton level, we confirmed that our approach is indeed optimal and extracts the true likelihood ratio over the entire relevant parameter space. The parton-level study served as a first benchmark for the novel morphing-aware approach. In the considered scenario, we found it to perform reasonably well but worse than the derivative learning method. At reco-level we produced numerically stable results, with a significant improvement over a rate measurement and a 1-dimensional histogram of the $WZ$-invariant mass. Our improved tool set will be the basis of a publicly available implementation as part of the next MadGraph/MadNIS release.

## Acknowledgments

We would like to thank Robert Schöfbeck and Dennis Schwarz, the Vienna Glühwein workshop 2023, Johann Brehmer, and the L-GATr team, without whom this study would not have been successful.

**Funding information**   This research is supported through the KISS consortium (05D2022) funded by the German Federal Ministry of Education and Research BMBF in the ErUM-Data action plan, by the Deutsche Forschungsgemeinschaft (DFG, German Research Foundation) under grant 396021762 – TRR 257: *Particle Physics Phenomenology after the Higgs Discovery*, and through Germany's Excellence Strategy EXC 2181/1 – 390900948 (the *Heidelberg STRUCTURES Excellence Cluster*). We would also like to thank the Baden-Württemberg Stiftung for financing through the program *Internationale Spitzenforschung*, project *Uncertainties - Teaching AI its Limits* (BWST_ISF2020-010). H.B. acknowledges support by the Alexander von Humboldt foundation. V.B. acknowledges financial support from the Grant No. ASFAE/2022/009 (Generalitat Valenciana and MCIN, NextGenerationEU PRTR-C17.I01).

## A   Assessing the learned likelihood

Since the actual likelihood is intractable (apart from toy models, where we can calculate it analytically), the error of the learned likelihood is also intractable. Nevertheless, it is possible to evaluate the quality of the learned likelihood.

### Uncertainties due to limited training data

As a first aspect, we can assess the uncertainties due to limited training data. This becomes especially important if we learn the likelihood ratio only at one point using derivative learning and then extrapolate from there. While this is a good approximation if the distributions at a different parameter point cover the same phase regions, this prescription necessarily breaks down for separated phase distributions. A similar issue can occur for the morphing-aware setup if interpolating between two benchmark points or if the likelihood is evaluated at a parameter point far from all benchmark points.

To assess in which parameter region the learned likelihood ratio gives precise results, we can learn the uncertainty due to limited training data in a specific phase-space region as part of

the network training. In general, there are multiple ways to implement this: e.g., via Bayesian neural networks [2, 46–50] or repulsive ensembles [2, 33, 34]. Here, we use the repulsive ensemble approach, but similar results are expected when using Bayesian neural networks.

A repulsive ensemble consists of a set of normal neural networks which are trained in parallel. The loss function is adapted to include a repulsive term which forces the networks to cover as much of their weight space as possible while still converging on the actual training objective. If the training sample is sparse in a certain phase-space region, it becomes easier for the networks to provide a reasonable fit. Due to the repulsive term, this results in a wider spread between the networks. Eventually, the average of the networks is used as the actual prediction of the ensemble, while their standard deviation is used to estimate the uncertainty due to limited training data.

If we use repulsive ensembles in the morphing-aware approach, we train one repulsive ensemble for every benchmark point. In the derivative approach, one repulsive ensemble is trained for each derivative w.r.t to the theory parameters. In both approaches, these ensembles are then used to build the total likelihood ratio (see Sec. 2). The estimated uncertainty of the total likelihood ratio can be obtained either

- by error propagation of the uncertainties of the individual ensemble, or

- by computing the ensemble of the total likelihood ratio combining the ensembles of trained networks, and then computing the standard deviation over the resulting sample.

In the second case, the selection of one network out of each ensemble can be randomized. Since the members of the ensemble are not ordered in any specific way, this choice does not make any difference in practice. If the number of ensemble members is large enough, both methods should give the same result. In the present work, we follow the second approach.

**Limit setting and empirical coverage**

As an additional diagnostic tool, we can evaluate the statistical properties of the learned likelihood. The full likelihood for a set of events $\{x_i\}$ is given by

$$p_{\text{full}}(\{x\}|\theta) = \text{Pois}(n|\mathcal{L}\sigma(\theta))\prod_i p(x_i|\theta), \tag{A.1}$$

consisting out of the the total rate term $\text{Pois}(n|\mathcal{L}\sigma(\theta))$ with the luminosity $\mathcal{L}$ and the kinematic likelihoods for each event $p(x_i|\theta)$. Here, $\text{Pois}(k|\lambda) = \lambda^k e^{-\lambda}/k!$ is the Poisson distribution. We denote the corresponding full likelihood ratio as $r_{\text{full}}(\{x\}|\theta, \theta_0)$.

Using the full likelihood, we construct the test statistics

$$q(\theta) = -2\log r_{\text{full}}(\{x\}|\theta, \hat{\theta}) = -2\left(\log r_{\text{full}}(\{x\}|\theta, \theta_0) - \log r_{\text{full}}(\{x\}|\hat{\theta}, \theta_0)\right), \tag{A.2}$$

where $\hat{\theta}$ is its minimum which we estimated via

$$\hat{\theta} = \underset{\theta}{\text{argmax}}\log r_{\text{full}}(\{x\}|\theta, \theta_1). \tag{A.3}$$

In the asymptotic limit, the distribution $p(q(\theta)|\theta)$ is a chi-squared distribution. We can then calculate the $p$-value

$$p_\theta = \int_{q_{\text{obs}}(\theta)}^{\infty} dq\, p(q(\theta)|\theta) = 1 - F_{\chi^2}(q_{\text{obs}}(\theta)|k), \tag{A.4}$$

which measures the confidence with which we can reject $\theta$. Here, $F_{\chi^2}(x|k)$ is the cumulative chi-squared distribution function with $k$ degrees of freedom. $q_{\text{obs}}(\theta)$ is the observed value for $q(\theta)$. Using this, the $\gamma$ confidence region is defined by all $\theta$ values for which $p_\theta < \gamma$.

As a consistency check for our learned likelihood ratio $r_\varphi(x|\theta, \theta_0)$, we can generate $n$ samples for a given $\theta$ and check for how many of the $n$ samples the true $\theta$ is contained in a given $\gamma$ confidence region. This defines the coverage

$$c_\gamma \equiv \left\langle \mathbb{1}\left(p_{\theta_0}(\{x\}) > 1 - \gamma\right)\right\rangle_{\{x\}}, \tag{A.5}$$

where $\mathbb{1}$ is the indicator function, which evaluates to one if the condition in the brackets is true and to zero otherwise. The average value is taken over the $n$ samples. The coverage is exactly $\gamma$ if the likelihood estimation is perfect. This means the true minimum is within the $\gamma$ confidence level in a fraction $\gamma$ of all cases. If the fraction is higher — meaning that the empirical $c_\gamma > \gamma$ —, our learned likelihood is conservative or underconfident. If it is lower — meaning that the empirical $c_\gamma < \gamma$ —, the learned likelihood is overconfident.

## B  Fractional-smearing datasets

This original dataset before fractional smearing is shown in the left panel of Fig. 13. This Figure shows the second derivatives of either the parton-level differential cross-section or the reco-level differential cross-section w.r.t. $\theta$ as a function of either $x$ for the reco-level points or $z$ for the parton-level points. The blue points show the parton-level distribution of events; the orange points, the reco-level distribution.

Looking at Fig. 13, the issue with the network training becomes immediately obvious. The parton-level events (blue circles) are smeared randomly to the left or right to obtain the reco-level training dataset (orange triangles). Since the event density in the high $x$ or $z$ region is low, this smearing can result in a random bias, making it very hard for the network to converge towards the true distribution.

This issue is alleviated by the fractional smearing procedure, as illustrated in the right panel of Fig. 13. Now every event with a high target is repeatedly smeared resulting in a distribution of smeared events (orange triangles) around each parton-level event (blue circles). This makes it easier for the network to learn the reco-level distribution.

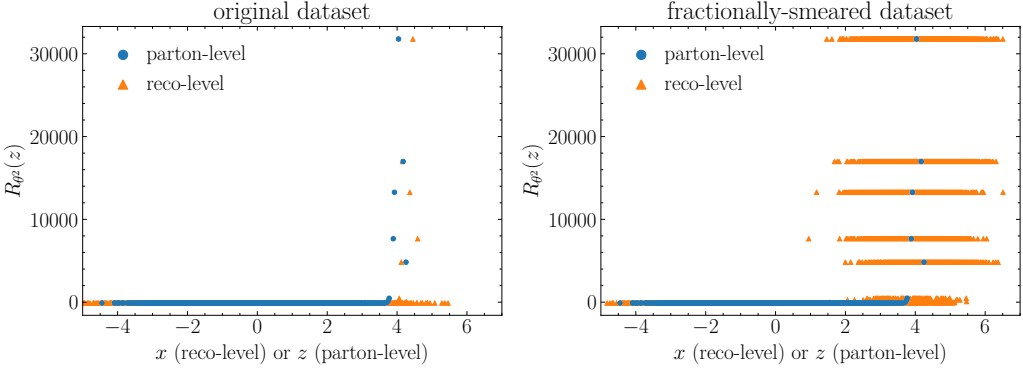

Figure 13: Left: Second derivative of the differential cross-section with respect to $\theta$ as a function of $x$ for the reco-level points or $z$ for the parton-level points. Shown is the parton-level (blue) and the reco-level (orange) distribution for the original dataset. Right: Same as left but for the fractionally-smeared dataset.

## C  Morphing-aware likelihood estimation for $pp \to W^{\pm}Z$

For the morphing-aware estimation of the $pp \to W^{\pm}Z$ likelihood at parton level, we choose the following basis points

$$
\begin{aligned}
&\theta_1 = (-4, 0, 0), &&\theta_2 = (4, 0, 0), &&\theta_3 = (0, -0.2, 0), \\
&\theta_4 = (0, 0.2, 0), &&\theta_5 = (0, 0, -0.2), &&\theta_6 = (0, 0, 0.2), \\
&\theta_7 = (-1.2, -0.09, 0), &&\theta_8 = (-1.2, 0, -0.09), &&\theta_9 = (0, -0.09, -0.09),
\end{aligned}
\tag{C.1}
$$

where $\theta = (c_{\Phi WB}, c_{WWW}, c_{\Phi q}^{(3)})$. We find that choosing basis points along the coordinate axis yields more stable results due to a simplification of the morphing matrix inversion.

Moreover, we choose a different training loss to the one used in Sec. 3. Instead of the BCE loss defined in Eq. 11, we use an MSE-based loss, which we find to yield more stable results. In particular, the loss reads

$$
\mathcal{L} = \left\langle \left[ r(z_{p,0}|\theta_i, \theta_0) - r_{\varphi}(x_0|\theta_i, \theta_0) \right]^2 + \left[ \frac{1}{r(z_{p,i}|\theta_i, \theta_0)} - \frac{1}{r_{\varphi}(x_i|\theta_i, \theta_0)} \right]^2 \right\rangle_{x_i, z_{p,i} \sim p(x|z_p)p(z_p|\theta_i)},
\tag{C.2}
$$

where the first term is only evaluated for events from the denominator hypothesis while the second term is only evaluated for events from the numerator hypothesis. The inversion of the ratio in the second term ensures convergence to the correct true ratio limit.

## D  Backgrounds

In the presence of backgrounds, the squared matrix element can be written in the form

$$
\mathcal{M}^2(z_p|\theta) = \mathcal{M}_{\text{sig}}^2(z_p|\theta) + \underbrace{2\text{Re}\left[ \mathcal{M}_{\text{sig}}(z_p|\theta)\mathcal{M}_{\text{bkg}}^*(z_p) \right]}_{\text{intf}} + \mathcal{M}_{\text{bkg}}^2(z_p),
\tag{D.1}
$$

where $\mathcal{M}_{\text{sig}}(z_p|\theta)$ is the matrix element of the signal process and $\mathcal{M}_{\text{bkg}}(z_p)$ is the matrix element of the background process, which does not depend on $\theta$.

If the signal and background do not have the same partonic final state, the interference term is zero. In this case, we can split up the differential reco-level cross-section,

$$
d\sigma(x|\theta) = d\sigma_{\text{sig}}(x|\theta) + d\sigma_{\text{bkg}}(x),
\tag{D.2}
$$

and also the likelihood,

$$
p(x|\theta) = \frac{\sigma_{\text{sig}}(\theta)}{\sigma_{\text{sig}}(\theta) + \sigma_{\text{bkg}}} p_{\text{sig}}(x|\theta) + \frac{\sigma_{\text{bkg}}}{\sigma_{\text{sig}}(\theta) + \sigma_{\text{bkg}}} p_{\text{bkg}}(x).
\tag{D.3}
$$

Then, we can write the likelihood ratio in the form

$$
r(x|\theta, \theta_0) = \frac{\sigma_{\text{sig}}(\theta_0) + \sigma_{\text{bkg}}}{\sigma_{\text{sig}}(\theta) + \sigma_{\text{bkg}}} \frac{1 + \frac{\sigma_{\text{sig}}(\theta)}{\sigma_{\text{bkg}}} \frac{p_{\text{sig}}(x|\theta_0)}{p_{\text{bkg}}(x)} \frac{p_{\text{sig}}(x|\theta)}{p_{\text{sig}}(x|\theta_0)}}{1 + \frac{\sigma_{\text{sig}}(\theta_0)}{\sigma_{\text{bkg}}} \frac{p_{\text{sig}}(x|\theta_0)}{p_{\text{bkg}}(x)}}.
\tag{D.4}
$$

We can now separately learn a background–signal classifier at the point $\theta_0$ to extract the ratio $p_{\text{sig}}(x|\theta_0)/p_{\text{bkg}}(x)$. Then, the signal-only likelihood ratio $p_{\text{sig}}(x|\theta)/p_{\text{sig}}(x|\theta_0)$ can be learned separately using the methods outlined in the main body of the text. A similar approach has already been applied in Ref. [13].

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
