# Peer review of "Advancing Tools for Simulation-Based Inference"

_SciPost Physics, doi:SciPost Phys. Core 8, 060 (2025)_

## Round 1 · Referee Report · Samuel Homiller (Referee 1) · 2025-5-21

Report

Thanks to the authors for their updates -- they've clarified all of my questions and addressed all my comments, and I am more than happy to recommend the paper for publication in it's present form.

Recommendation

Publish (easily meets expectations and criteria for this Journal; among top 50%)

---

## Round 1 · Author Response

Dear Editor and Referee,

We thank the referee for his time, careful consideration, and the positive evaluation of our manuscript. We list below the changes we have made concerning the helpful suggestions.

*My primary questions/comments are with regards to understanding the scope of these improvements, and the possibilities for future applications. While the factorization in Eq. (19) is not limited to only SMEFT examples, this form doesn't necessarily hold when moving beyond leading order. In fact, for exactly the WZ process chosen as an application in this paper, the NLO corrections spoil this form: in contrast to many other diboson/Higgs processes where an NLO K-factor is sufficient, WZ production has a radiation zero that is particularly sensitive to anomalous couplings but which is spoiled by QCD radiation (see arXiv:1909.11576), leading to corrections that change as a function of both the kinematics and the theory parameters.*

We agree that for WZ production the size of the NLO corrections is dependent on both the kinematics and the entering Wilson coefficients and now also mention this in the text at the beginning of Section 4. The factorization of Eq.(19) still holds beyond leading order. It is simply based on the polynomial structure of the matrix element which is still present beyond leading order. We now also comment on this below the equation.

*The authors apply a jet veto in their reconstruction level analysis, so I believe their results in Section 4 are unaffected by this, but perhaps the authors can comment on whether their techniques can be applied to full NLO generation (since this at least in principle now straightforward with MG5 and SMEFT@NLO)? A parton-level analysis is probably ill-defined, but it seems to me that the morphing-aware learning would still help in getting stable results at the reconstruction level, at least compared to derivative learning where the expansion in the theory parameters may be very unstable? Fractional smearing also seems as though it would be especially useful in this case, since it would help interpolate to the amplitude with the additional hard parton. If nothing else, perhaps the authors can comment on whether their methods would provide a reasonable, conservative estimate of the likelihood in the case the factorization assumed at parton level breaks down?*

Without explicitly doing the full NLO analysis, it is hard to predict the stability of the derivative-learning approach. We, however, agree that morphing-aware likelihood learning and fractional smearing would help alleviating potential instabilites. We now comment on this at the beginning of section 4.2.

*- Can the authors elaborate a bit on why the exact form of the loss in Eq. (17) is important? What about the problem changes if a different exponent or computing the MSE of log r as opposed to r is used instead that doesn't result in convergence?*

We added an explanatory sentence.

*- Can the authors briefly comment why L-GATr only yields marginal improvements at parton level, but significant ones at reconstruction level?*

We added an explanatory sentence.

We also fixed the minor issues apart from the "small typo below Eq. (38)". For us, the sentence is correct as it stands.

For convenience, we marked all changes in blue.

We hope that with these changes, our article can now be accepted for publication in its present form.

Sincerely,

H. Bahl, V. Bresó-Pla, G. Di Crescenzo, T. Plehn

---

## Editorial Decision

published